# Estimated Treatment Effects of Tight Glycaemic Targets in Mild Gestational Diabetes Mellitus: A Multiple Cut-Off Regression Discontinuity Study Design

**DOI:** 10.3390/ijerph17217725

**Published:** 2020-10-22

**Authors:** David Song, James C Hurley, Maryanne Lia

**Affiliations:** 1Internal Medicine Service, Ballarat Base Hospital, Ballarat, Victoria 3350, Australia; hurleyjc@unimelb.edu.au; 2Dorevitch Pathology, Chemical Pathology, Melbourne, Victoria 3084, Australia; 3School of Medicine, Deakin University, Geelong, Victoria 3217, Australia; 4Department of Rural Health|Melbourne Medical School, University of Melbourne, Ballarat, Victoria 3350, Australia; 5Peninsula Health, Melbourne, Victoria 3199, Australia; maryanne.lia94@gmail.com

**Keywords:** gestational diabetes mellitus, quasi-experimental design, regression discontinuity

## Abstract

*Background*: We investigated the treatment effects of tight glycaemic targets in a population universally screened according to the International Association of Diabetes and Pregnant Study Groups (IADPSG)/World Health Organisation (WHO) gestational diabetes mellitus (GDM) guidelines. As yet there, have been no randomized control trials evaluating the effectiveness of treatment of mild GDM diagnosed under the IADPSG/WHO diagnostic thresholds. We hypothesize that tight glycaemic control in pregnant women diagnosed with GDM will result in similar clinical outcomes to women just below the diagnostic thresholds. *Methods*: A multiple cut-off regression discontinuity study design in a retrospective observational cohort undergoing oral glucose tolerance tests (OGTT) (*n* = 1178). Treatment targets for women with GDM were: fasting capillary blood glucose (CBG) of ≤5.0 mmol/L and the 2-h post-prandial CBG of ≤6.7 mmol/L. Regression discontinuity study designs estimate treatment effects by comparing outcomes between a treated group to a counterfactual group just below the diagnostic thresholds with the assumption that covariates are similar. The counterfactual group was selected based on a composite score based on OGTT plasma glucose categories. *Results*: Women treated for GDM had lower rates of newborns large for gestational age (LGA), 4.6% versus those just below diagnostic thresholds 12.6%, relative risk 0.37 (95% CI, 0.16–0.85); and reduced caesarean section rates, 32.2% versus 43.0%, relative risk 0.75 (95% CI, 0.56–1.01). This was at the expense of increases in induced deliveries, 61.8% versus 39.3%, relative risk 1.57 (95% CI, 1.18–1.9); notations of neonatal hypoglycaemia, 15.8% versus 5.9%, relative risk 2.66 (95% CI, 1.23–5.73); and high insulin usage 61.1%. The subgroup analysis suggested that treatment of women with GDM with BMI ≥30 kg/m^2^ drove the reduction in caesarean section rates: 32.9% versus 55.9%, relative risk 0.59 (95%CI, 0.4–0.87). Linear regression interaction term effects between non-GDM and treated GDM were significant for LGA newborns (*p* = 0.001) and caesarean sections (*p* = 0.015). *Conclusions*: Tight glycaemic targets reduced rates of LGA newborns and caesarean sections compared to a counterfactual group just below the diagnostic thresholds albeit at the expense of increased rates of neonatal hypoglycaemia, induced deliveries, and high insulin usage.

## 1. Introduction

The International Association of the Diabetes and Pregnancy Study Groups (IADPSG) [1] gestational diabetes mellitus [1] (GDM) diagnostic thresholds, endorsed by the World Health Organisation (WHO) [2], are not universally accepted [3]. The ongoing lack of consensus reflect concerns of the increasing costs of higher GDM prevalence using the one-step universal screening with lower oral glucose tolerance test (OGTT) cut-offs and the limited evidence on the clinical effectiveness of treating milder disease [4]. Indeed, there is disagreement between those suggesting new randomised controlled trials (RCT) are required and those who believe existing RCTs, which did not use the current IADPSG/WHO guidelines, with observational studies, are sufficient [4,5]. In rural communities, access to specialist care may be limited and the clinical impact of increased GDM prevalence by lowering diagnostic thresholds in these cohorts are yet to be clarified.

In centres where the IADPSG/WHO GDM guidelines have become the standard of care, RCTs comparing outcomes of new to old guidelines may be ethically difficult to perform. Moreover, RCTs for mild GDM can be challenging as the act of randomisation may reduce treatment effects by affecting the behaviour of control groups [6]. This is salient in GDM, as the primary interventions are behavioural modification: dietary and lifestyle.

Variations in obstetric practises challenge the external validity of RCTs. For instance, the reduction of caesarean section rates is purported to be a driver of the cost-effectiveness of treating GDM [7]. Since caesarean section rates differ widely around the world [8], results from multi-centre RCTs may not have local applicability.

We propose a new method of estimating GDM treatment effects in local observational cohorts. GDM is unusual as there is no natural threshold differentiating disease and non-disease populations. There is a continuous relationship between OGTT plasma glucose levels and clinical outcomes [1]. We use a multiple cut-off regression discontinuity study design analysing the clinical outcomes around the IADPSG/WHO diagnostic thresholds to estimate treatment effects, utilising the continuous relationship of OGTT plasma glucose to GDM outcomes.

Regression discontinuity studies use variations in assignment methods to emulate random treatment assignment, analogous to RCTs [9]. For instance, when diagnostic tests are used to assign patients, random analytical/pre-analytical variations of test results around the threshold acts to quasi-randomise patients to an intervention [9]. This quasi-random allocation of patients based on OGTT is pertinent given the poor reproducibility of OGTT for individuals [10]; nonetheless, there are robust correlations of GDM outcomes to OGTT plasma glucose for a population [11].

## 2. Methods

We received Ballarat Health Services and St John of God Healthcare Human Research Ethics Committee approval for this study (LNR/18/BHSSJOG/16).

### 2.1. Study Population and Sampling Design

This single-centre retrospective cohort study searched records of women birthing at Ballarat Health Services (BHS) between 1 January 2017 and 31 December 2017. BHS is a regional public hospital in the state of Victoria, Australia with a population of approximately 100,000. BHS delivers approximately 70% of births in Ballarat. Additionally, women with GDM requiring intensification of therapy with medications are referred to the antenatal clinic from up to 150 km away.

We used the Australasian Diabetes in Pregnancy Society [12] (ADIPS) guidelines, which recommend single-step universal screening for GDM by 24–28 weeks gestation using 75 g anhydrous glucose oral glucose tolerance test. The ADIPS guidelines follow IADPSG GDM thresholds. A GDM diagnosis was based on one or more results meeting the following plasma glucose levels: fasting plasma glucose 5.0–6.9 mmol/L, 1-h ≥10.0 mmol/L, or 2-h 8.5–11.0 mmol/L. The pre-analytical and analytical OGTT methods were described in a study by Song D et al. [13]. Women diagnosed as having GDM had OGTT at any time during pregnancy consistent with local practises.

Inclusion criteria were all singleton births at BHS between 1 January 2017 and 31 December 2017 with OGTTs.

Exclusion criteria were women with overt diabetes mellitus as defined by OGTT levels fasting plasma glucose ≥7.0 mmol/L and 2-h plasma glucose of ≥11.1 mmol/L; and women with GDM who did not receive GDM antenatal interventions at BHS. For women below the GDM diagnostic thresholds, we excluded those with incomplete OGTTs, missing 1 h and 2 h plasma glucose levels; and those with OGTTs before 24 or after 28 weeks.

Data were collected from the initial antenatal clinic visit until after delivery. The hospital’s electronic records were searched for OGTT, demographic data (age, ethnicity, gravida, parity, body mass index (kg/m^2^) at first presentation, foetal sex), maternal outcomes (induced deliveries, caesarean sections, gestation age of delivery, diabetes mellitus diagnosis, medication use), and foetal outcomes (delivery weight, foetal length, neonatal hypoglycaemia, shoulder dystocia).

Large for gestational age newborn (LGA) and small for gestational age newborn (SGA) was defined as a birth weight greater than the 90th and less than the 10th percentile, for gestational age respectively, using age based standardised charts [14]. Gestational age was based on either ultrasound reports of gestational age or the last menstrual period. Neonatal hypoglycaemia was based on documentation in the database.

### 2.2. Interventions

Women with GDM typically received initial group education by the diabetes educator and the dietician. Women were advised to test capillary blood glucose (CBG) at fasting and 2 h after the beginning of each significant meal. In subsequent appointments GDM women were given individualised advice on the recommended weight gain in pregnancy and diet, as appropriate. Recommended weight gain suggestions, for women with or without GDM, in pregnancy were based on the Institute of Medicine guidelines [15], except for those with BMI >35, where the local consensus was for minimal further weight gain.

The standard management for women with GDM included attendance at the multi-disciplinary antenatal clinic one week after the initial education, this clinic included endocrinologists, obstetricians, midwives, anaesthetists, dieticians, and diabetes educators. For those diagnosed early in pregnancy, women typically attended four weekly appointments until 28–30 weeks; thereafter, women were typically seen two-weekly. Most women were encouraged to self-titrate insulin/metformin doses, within strict parameters, to reach target CBG levels.

### 2.3. Treatment Targets

Treatment targets and thresholds for intensification were consistent with ADIPS suggestions [12], which were fasting capillary blood glucose ≤5.0 mmol/L and 2-h CBG after commencing meal ≤6.7 mmol/L. The decision to consider intensification of treatment were based on whether there were two or more elevated CBG levels at a given time within one week, CBG pattern, ultrasound reports on estimated foetal weight/abdominal circumference, and women’s preferences.

The basal insulin used was Isophane (Novo Nordisk, Denmark) prescribed nocte. The bolus insulin used was Aspart (Novo Nordisk, Denmark). The metformin used was the modified-release formulation.

### 2.4. GDM Classification and Composite Score

Patients were assigned to GDM or non-GDM groups as per IADPSG/WHO guidelines. To further categorise patients, all were given a Hyperglycaemia and Adverse Pregnancy Outcomes (HAPO) glucose composite score. Each patient was categorized according to HAPO glucose categories [13]:

Fasting plasma glucose: category 1, <4.2 mmol/L; category 2, 4.2 to 4.4 mmol/L; category 3, 4.5 to 4.7 mmol/L; category 4, 4.8 to 4.9 mmol/L, category 5, 5.0 to 5.2 mmol/L; category 6, 5.3 to 5.5 mmol/L; category 7, 5.6 to 7.0 mmol/L.

For 1 h plasma glucose: category 1, ≤5.8 mmol/L; category 2, 5.9 to 7.3 mmol/L; category 3, 7.4 to 8.6 mmol/L; category 4, 8.7 to 9.5 mmol/L; category 5, 9.7 to 10.7 mmol/L; category 6, 10.8 to 11.7 mmol/L; category 7, ≥11.8 mmol/L.

For 2 h plasma glucose: category 1, ≤5.8 mmol/L; category 2, 5.9 to 7.3 mmol/L; category 3, 7.4 to 8.6 mmol/L; category 4, 8.7 to 9.5 mmol/L; category 5, 9.7 to 10.7 mmol/L; category 6, 10.8 to 11.7 mmol/L; category 7, ≥11.8 mmol/L.

The range of fasting plasma glucose levels in a population was narrower than the 1 h and 2 h levels, each patient’s OGTT results were scaled according to a measure with equivalent risks of adverse outcomes—the HAPO study’s glucose categories (between 1 to 7) [13]. The HAPO glucose composite score for each patient was the sum of scores of maternal plasma glucose exposure at each OGTT time point. The HAPO glucose composite score serves as the dependent variable used in linear regression analysis and the bandwidth selection of the counterfactual group.

The range of possible HAPO glucose composite scores was from 3 (1 + 1 + 1) to the maximum of 21 (7 + 7 + 7). As the GDM diagnostic thresholds intersects category 5 at all OGTT time points, for GDM patients the theoretical minimum HAPO glucose composite score was 7 (5 + 1 + 1). For non-GDM patients, the maximum possible HAPO glucose composite score was 15 (5 + 5 + 5). Between GDM and non-GDM patients, there was a possible overlap of HAPO glucose composite scores from 7 to 15.

### 2.5. Study Design

Our study is a regression discontinuity design structured on a composite score derived as a result of OGTT’s multiple cut-offs. The composite score generates an overlap between GDM and non-GDM patients.

Treatment effects for mild GDM were estimated by first, comparing outcomes of GDM patients to a counterfactual group not meeting the diagnostic thresholds for GDM. Bandwidth selection for the counterfactual group was selected on the basis of HAPO glucose composite scores where it also selects at least 95% of women with GDM. Second, as obesity is a major independent contributor to clinical adverse GDM outcomes of LGA and caesarean sections [16], we performed a subgroup analysis of patients based on BMI: <30 or ≥30 kg/m^2^. Third, linear regression was applied to LGA and caesarean sections, as they are known to have linear relationships to maternal glucose levels [13].

For the linear regression, we plotted caesarean section rates and LGA versus HAPO glucose composite scores. Where there were fewer numbers of non-GDM patients, they were pooled for HAPO glucose composite scores: 9 to 10, and 11 to 14. GDM patients were similarly pooled at the extremes of the HAPO glucose composite score categories: 7 to 10, 15 to 19, for the purposes of the linear regressions. Where there were no LGA events in a HAPO glucose composite category score, we pooled patients with the adjacent score. The unpooled LGA events across HAPO glucose composite category scores are disclosed in the Appendix A.

### 2.6. Statistical Methods

The statistical significance for baseline demographic parametric data was calculated using non-paired Student *t*-tests and non-parametric baseline data was calculated using the Mann–Whitney test. Statistical significance for maternal and neonatal outcomes for proportions was calculated using the χ^2^ test; for parametric outcome data we used unpaired student *t*-test. The non-parametric bootstrap method was used to determine the 2.5, 97.5 percentile reference interval for OGTT fasting glucose [17]. Clinical outcomes were considered statistically significant at a *p* value of <0.05. A linear regression of LGA and caesarean sections versus HAPO glucose composite score was undertaken and the statistical significance of interaction term effects was determined. Statistics were analysed using Minitab 18, State College, PA, USA; and STATA 15, StataCorp. 2015, College Station, TX, USA.

## 3. Results

### 3.1. Counterfactual Group

Among 1505 women delivering at BHS, 1174 had complete OGTTs (Figure 1). There were 162 patients with OGTT above IADPSG/WHO GDM thresholds. After excluding overt diabetes and those patients who did not attend BHS GDM clinic, there were 152 GDM patients. Between 24–28 weeks gestation there were 888 patients after removal of those above diagnostic thresholds or with incomplete OGTT. There were 135 patients with HAPO glucose composite scores ≥9, which are here defined as the counterfactual group as more than 95% of GDM patients, except five, had HAPO glucose composite score ≥9 (Figure 1). The counterfactual group represented the highest of HAPO composite glucose scores in the non-GDM group.

The lower 2.5th percentile and the upper 97.5th percentile for OGTT fasting plasma glucose (*n* = 1178) were 3.7 mmol/L and 5.3 mmol/L, respectively.

Baseline characteristics of all patients with OGTTs, GDM, and the Counterfactual group are described in Table 1.

### 3.2. The Management of GDM Patients

Almost two-thirds of patients (65.7%), Table 2, required pharmaceutical intervention to meet treatment targets. The median basal dose was 16 units, (interquartile range 8 to 26), with the maximum dose of 190 units. Bolus insulin was used in 43 patients (28.3% of all GDM patients); median total daily dose was 10 units, (interquartile range 4–18) and maximum dose 230 units. Insulin with metformin was used in 7.2% (*n* = 11) patients.

### 3.3. Maternal Outcomes

Compared to the counterfactual group below the diagnostic thresholds (4), women with GDM delivered earlier (*p* < 0.001) had 57% higher rates of induced deliveries (Table 3). There was a non-significant trend (*p* = 0.06) for lower risk of caesarean delivery, however the interaction effects in the linear regression were statistically significant (*p* = 0.015; Figure 2).

The study centre’s overall rate of caesarean births for women with OGTT at BHS was 27.8% (*n* = 328/1178).

### 3.4. Neonatal Outcomes

GDM treated neonates had lower mean birth weight, 3224 g versus 3516 g (*p* < 0.001), and lower neonatal mean BMI 13.4 versus 14.0 kg/m^2^ (*p* = 0.001) (Table 4). GDM patients had lower rates of LGA (4.6% versus 12.6%) and RR 0.37 (95%CI, 0.16–0.85). Increases in SGA rates did not reach statistical significance. We note that five of six patients with SGA were from GDM women managed with diet/lifestyle only.

Neonatal hypoglycaemia increased in GDM patients 15.8% versus 5.9%, RR 2.66 (95% CI, 1.23–5.73) but with no difference in the admissions to the special care nursery (Table 4).

### 3.5. Subgroup Analysis

The women with GDM and BMI ≥30 had reduced absolute caesarean section rates (32.9% versus 55.9%), RR 0.59 (95% CI, 0.4–0.87); and primary caesarean section rates (26.4% versus 45.8%), RR 0.58 (95%CI, 0.35–0.94) when compared to the counterfactual group with BMI ≥30. For women with BMI <30, there were no statistically significant differences in total and primary caesarean section rates (Table 4).

### 3.6. Linear Regression

Figure 2 demonstrates a change in the gradient suggesting discontinuity for GDM treated compared to non-GDM patients for LGA and caesarean sections. Linear regression interaction term effects were significant for LGA (*p* = 0.001) and caesarean section (*p* = 0.015).

## 4. Discussion

### 4.1. Study Design

Regression discontinuity studies assume that there is a continuous relationship between outcome variables [9] such as rates of LGA, caesarean sections and the dependent variable, the HAPO glucose composite scores. We demonstrate a linear continuous relationship between LGA and caesarean sections and HAPO glucose composite scores in non-GDM patients (Figure 2).

Regression discontinuity studies typically estimate treatment effects proximate to the diagnostic cut-off [9]. However, the diagnosis of GDM is unusual in having multiple diagnostic cut-offs at OGTT fasting, 1 h, and 2 h plasma glucose. Furthermore, the heterogeneity of OGTT plasma glucose trajectories with multiple cut-offs resulted in overlaps in the HAPO glucose composite scores, between GDM and non-GDM patients. This allowed inference of treatment effects to wider bandwidths than other forms of regression discontinuity designs. Nevertheless, the counterfactuals represent those with milder GDM.

### 4.2. Population Characteristics

Our GDM prevalence of 13.5% was similar to Australian HAPO centres: 12.4% to 15.3% [18]. GDM patients compared to the counterfactual group had higher BMI. Expectedly, as OGTT were used to determine diagnosis, GDM patients had higher OGTT plasma glucose levels. As covariates associated with adverse outcomes were higher for GDM patients, the interventions had to overcome these selection biases to show improved outcomes.

Our centre’s overall caesarean section rate of 27.8% was similar to the Organisation for Economic Co-operation and Development rate of 27.9% [19]; but below the Australian rate of 34% [19].

Presumably, higher maternal glucose levels are the driver of adverse clinical outcomes in GDM [11] and that moderating elevated maternal glucose levels improves outcomes. Therefore, it is important to discuss clinical outcomes with the understanding of treatment targets and thresholds for treatment intensification. ADIPS [12] suggested treatment targets were fasting CBG ≤5.0 mmol/L and the post-prandial ≤6.7 mmol/L. Ostensibly the aim was to have maternal CBG levels within the reference range of a healthy population [9]. This is tighter than RCT CBG targets in the Australian Carbohydrate Intolerance Study (ACHOIS) [20]: fasting ≤5.5 mmol/L, and 2 h post-prandial of ≤7.0 mmol/; and the Maternal-Fetal Medicine Unit study (MFMU) [21]: fasting ≤5.3 mmol/L, and 2 h post-prandial of ≤6.7 mmol/L.

The threshold for intensifying treatment will also impact medication usage. ADIPS [9] suggested intensification if two out of seven results (29%) were above target; in comparison, the MFMU study [21] intensified treatment if more than 50% of CBG were above target. Glucometer imprecision will affect treatment intensification; the median coefficient of variation (CV) among of 18 commercial glucometers in recent survey was 9.3% [22]. To illustrate this point, take a theoretical woman with ‘true’ consistent fasting CBG levels within recommended levels at 5.0 mmol/L, but due to glucometer imprecision 95% (2 sd or 2CV) of results would be between 4.1 to 5.9 mmol/L, of which half would be >5.0 mmol/L. Therefore, it is probable that more than 29% of fasting CBG will be above 5.0 mmol/L, necessitating treatment intensification.

Notwithstanding glucometer imprecision issues, our population’s upper reference interval (97.5th percentile) of OGTT fasting plasma glucose was 5.3 mmol/L. Our OGTTs were collected using sodium fluoride collected at room temperature and batched centrifuged [13]. This pre-analytical method has been recently documented, via the process of glycolysis, to lower reported OGTT fasting glucose levels by approximately 10% [13,23]. Therefore, the adjusted lower and upper reference level of fasting glucose should be 10% higher—approximately 4.1 to 5.8 mmol/L. A fasting glucose target of 5.0 mmol/L is well within our population’s reference intervals.

### 4.3. Medication Usage

Of women with GDM, 61.1% were treated with insulin (Table 2). Nearly all of those on insulin were on Isophane insulin, aimed to treat elevated fasting CBG. Our usage of insulin is higher than reported in randomised control studies: ACHOIS [20] of 20%, and MFMU [21] of 7.6%. ADIPS treatment targets were tighter and so were the thresholds for the intensification of therapy. Moreover, as the effective treatment target for the fasting CBG level was well within our population’s reference intervals the extent of basal insulin usage was unsurprising.

### 4.4. Clinical Outcomes

Treated GDM patients had a relative risk of LGA of 0.37, reducing the absolute risk from 12.6 to 4.6%. The RCTs, ACHOIS [20] and MFMU [21], reported a decrease in LGA rates from 22% to 13%, 13% to 7.1%, respectively. As our CBG targets were lower it was not surprising that our rates of foetal overgrowth were also lower. The Pedersen hypothesis that states that maternal hyperglycaemia results in foetal hyperinsulinemia, glucose utilization, and subsequent foetal adiposity [24]. The HAPO study demonstrated the continuous linear associations between maternal glycaemia and cord c-peptide levels and rates of LGA [11]. It follows that lowering maternal glycaemia should lower rates of LGA and tight maternal glycaemic control should result in lower rates of LGA. Our study further supports the Pedersen hypothesis by demonstrating that tight glycaemic targets lower rates of LGA, even when compared to patients just below the diagnostic thresholds.

The relative risk for caesarean sections for GDM as compared to the counterfactual group was 0.75, the absolute risk reduced from 43.0% to 32.3%. Subgroup analysis suggests the reduction of caesarean section rates were driven by women with BMI ≥30, where the relative risk was 0.58 compared to counterfactual group with BMI ≥30. The absolute risk decreased from 55.9% to 32.9%. In contrast, GDM women with BMI <30 did not demonstrate a reduction in caesarean section rates. Our study suggests the treatment response upon caesarean section rates is modified by obesity.

The two randomised led trials of GDM had shown contrasting caesarean section rates. ACHOIS [20], showed no effect of treatment on caesarean section rates, but in the MFMU study [21], treatment resulted in a relative risk of 0.79 (*p* = 0.02). Although these studies differed in treatment targets and selection of patients, our BMI subgroup analysis may partly explain the differences in caesarean section outcomes. The baseline BMI of patients differed between these two RCTs, the trial not showing caesarean rate reduction had lower median BMI of 26.8 [20], the other had a mean BMI of 30.1 kg/m [2,21].

Apart from tighter treatment targets, another factor that may have contributed to the larger absolute and relative caesarean risk reduction seen in our study compared to the RCTs is the marked differences in the baseline risks of caesarean sections. Our counterfactual group had caesarean section rate of 43.0%, driven primarily by women with BMI ≥30; this is in the context of the overall study centre rate of 27.8%. In contrast, the RCTs had rather muted untreated GDM (control) caesarean section rates of 32% and 33.8%. This is in the context of national caesarean section rates in Australia (2004) [25] of 29.4% and the United States (2005) [26] of 30.3%. We note the established linear correlation of caesarean section rates with increasing maternal plasma glucose [11]. Therefore, in women with very high maternal plasma glucose levels, as in untreated GDM, we would have expected much higher caesarean section rates than the centre or national levels. In our study, obesity was a strong driver of higher caesarean section rates; and our rates for the counterfactual group are consistent with the odds ratios seen in a meta-analysis investigating obesity’s impact on caesarean sections: 2.05 and 2.89 for obese and severely obese women, respectively [27].

The HAPO study demonstrated that both maternal BMI and glycaemia have strong independent associations with caesarean section rates [16]. However, the HAPO study could not conclude that there was a causal relationship between increased maternal BMI and caesarean sections rates [16]. Our subgroup analysis of patients based on BMI may suggest tight glycaemic targets in women with BMI <30 kg/m^2^ does not reduce the risk of caesarean sections. As caesarean section rates decreased in women with GDM with BMI ≥30 kg/m^2^, it is obesity coupled with maternal glycaemia that are possibly the important factors in determining rates of caesarean sections.

In RCTs, the act of randomisation and patient selection processes may have recruited women with better baseline prognoses relative to the target population [28]. RCTs select for particular types of willing participants [28]. Moreover, inclusion in a study may have affected the behaviour of patients, especially as the primary interventions in GDM are behavioural change. This may have resulted in RCTs lowering adverse outcomes in comparator controls thereby attenuating the treatment response.

As our counterfactual group represents those borderline for GDM, we likely understated the overall treatment response. If the counterfactual group were to represent untreated GDM, mild and severe, we would expect higher baseline adverse outcomes. Linear regression (Figure 2) visualises treatment effects beyond mild OGTT maternal plasma glucose levels.

GDM women had the relative risk of neonatal hypoglycaemia of 2.66. This may be the result of detection bias. The local protocol mandates screening for neonatal hypoglycaemia of all GDM women and neonates with a birth weight >4000 g. In a study of neonates with no risk factors of hypoglycaemia, close serial monitoring within the 24 h showed a neonatal hypoglycaemia rate, as defined by <2.6 mmol/L, of 14%, which is similar to our rate of 15.8% [29].

Increased rates of induced deliveries and consequently lower birth gestational age of GDM patients is perhaps due to clinicians’ perception of higher risk for GDM women.

### 4.5. Limitations and Strengths

Our method may not be suitable for centres using the currently recommended citrate tubes for OGTT plasma glucose. Samples using citrate tubes would report fasting glucose levels of approximately 10% higher [23]. The HAPO glucose categories will no longer represent equivalent risks of adverse outcomes, as the HAPO study did not use citrate tubes [13]. Furthermore, quasi-randomisation partially relies on variations in pre-analytical handling, which would be reduced with citrate tubes.

Our study has limitations. As a retrospective design we did not have complete data on GDM risk factors such as family history of diabetes mellitus or smoking, in addition to data on maternal weight gain, and maternal hypertension. We did not record adherence to therapy with records of CBG but the high medication usage rate indicate that our GDM patients were treated to ADIPS suggestions.

As a single centre study design with a predominately Caucasian population (Table 1) our results may not be applicable to all centres. Obstetric practices such as rates of caesarean section may vary between and within countries [8] and therefore our risk reduction of caesarean sections may not be able to be extrapolated.

LGA/SGA outcomes relied on age standardised growth charts not adjusted for ethnic composition. However, as our population was predominately Caucasian there would have been marginal impact.

Our study may have been underpowered to determine differences in incidence rates for some less common end points such as shoulder dystocia.

We expect the loss of patients by birthing at other centres to be rare, as our hospital is a referral centre for high-risk pregnancies for smaller regional hospitals. The closest hospital able to deliver high-risk pregnancies is more than 90 km from BHS. We did not see women expected to deliver at the local private hospital.

Patients were analysed on an intention-to-treat basis irrespective of the adherence to therapy. Since our patients were treated contemporaneously and by the same group of health professionals, our study is unlikely to be confounded by evolving medical practises.

It is our view that our study design is particularly suited to the study of GDM. This method, which can be performed in retrospective and prospective cohorts, will allow other centres with differing populations and obstetric practices to assess the local effectiveness of GDM treatments. Furthermore, this quasi-experimental design may be suitable to evaluate the effects of GDM interventions on childhood adiposity and glucose tolerance.

## 5. Conclusions

Treatment of GDM women screened using IADPSG/WHO guidelines with tight treatment target reduces LGA outcomes, and caesarean section rates when compared to non-GDM patients just below the diagnostic thresholds. The treatment response upon caesarean section rates appear to be modified by obesity with GDM women with BMI ≥30 kg/m^2^ having reductions in rates, but there was no suggested benefit in GDM women with BMI <30 kg/m^2^. This is at the expense of slightly earlier gestational age at delivery, and increases in induced deliveries, notations of neonatal hypoglycaemia, and medication usage.

We believe this multiple cut-off regression discontinuity study design is an ideal method to investigate the treatment effects upon mild GDM when RCTs are impractical. This design is uniquely suitable for the investigation for GDM due to the unusual feature of GDM: a continuous linear relationship between maternal plasma glucose and clinical outcomes.

## Figures and Tables

**Figure 1 ijerph-17-07725-f001:**
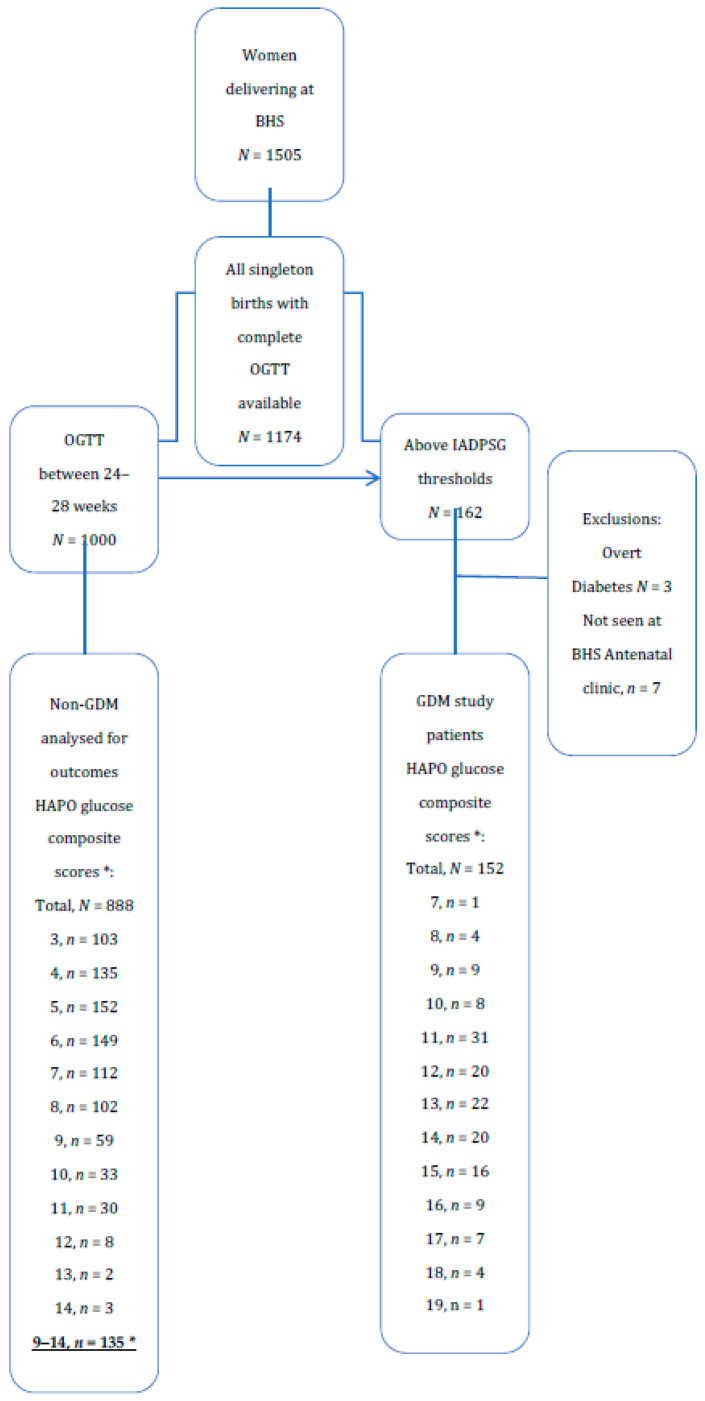
OGTT score and GDM diagnosis in women audited. * HAPO glucose composite scores are the sum of HAPO glucose scores at OGTT, fasting, 1 h, and 2 h.

**Figure 2 ijerph-17-07725-f002:**
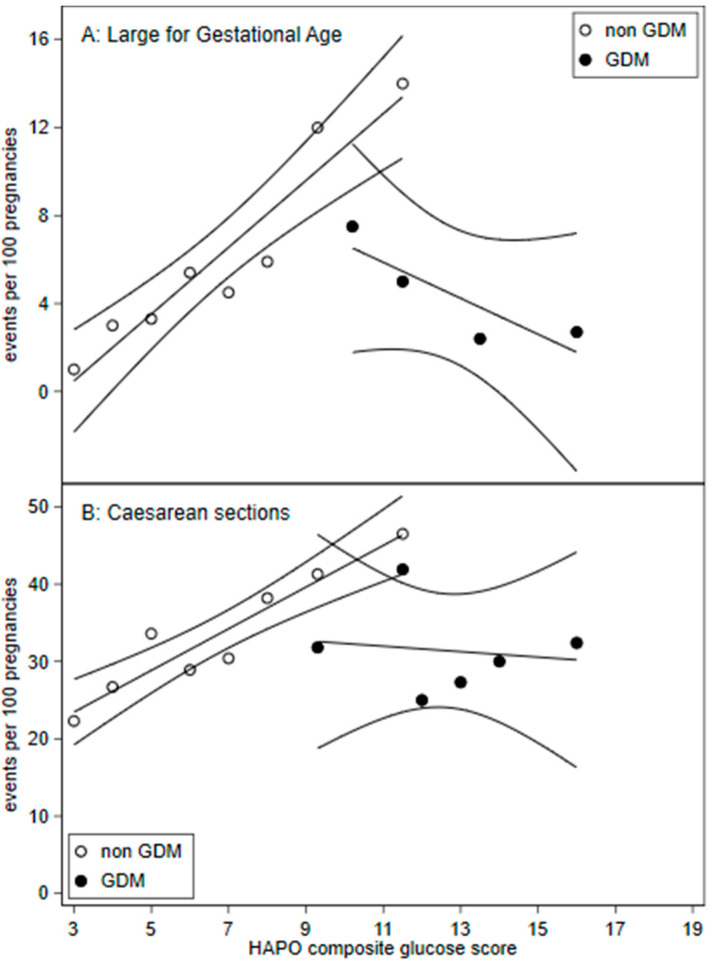
Frequency of (**A**) large for gestation age (birth weight >90th percentile) and (**B**) caesarean sections across HAPO glucose composite scores with 95% confidence intervals. HAPO glucose composite scores were calculated by adding the HAPO glucose categories for fasting, 1 h, and 2 h plasma glucose. A. Large for gestational age: GDM HAPO glucose composite score were pooled for scores: 7 to 11, 13 to 14, and 15 to 19.

**Table 1 ijerph-17-07725-t001:** Baseline characteristics of the study population.

Variable	GDM *n* = 152	Counterfactual Group *n* = 135	All Patients with OGTTs *n* = 1178
Age (years)	30.5 ± 5.5	30.4 ± 5.3	29.1 ± 5.4
BMI (kg/m^2)^	31.4 ± 8.2 *	29.5 ± 7.3	27.7 ± 7.0
Ethnicity%			
Caucasian	86	90	91
South Asian	6	4	3
East Asian	3	4	2
Other	5	2	3
Gravida	2.8 ± 1.7 ^ns^	2.6 ± 1.6	2.6 ±2.3
Parity ^§^	1.1 ± 1.2 ^ns^	1.1 ± 1.1	0.97 ± 1.1
OGTT (mmol/L)			
Fasting glucose	5.0 ± 0.4 **	4.5 ± 0.3	4.3 ±0.5
1 h	9.3 ± 1.6 **	8.5 ± 0.9	7.0 ±1.8
2 h	7.5 ± 1.5 **	7.0 ± 0.9	5.8 ±1.4
HAPO glucose composite score ^¶,#^	13 (11–13) *	10 (9–11)	6 (5–9)

Values are mean ± standard deviation unless otherwise specified. ^#^ Median and Interquartile range. *p* values as compared to the counterfactual group. ns, non-significant; * *p* 0.001 to <0.05; ** *p* < 0.001; Non-paired student *t*-test used to compare continuous variables and Mann-Whitney test used to compare non-parametric categorical variables. ^§^ Adjusted for parity at first presentation. ^¶^ HAPO glucose composite scores are the sum of HAPO glucose categories at OGTT, fasting, 1 h, and 2 h.

**Table 2 ijerph-17-07725-t002:** Interventions reported for women diagnosed with GDM.

Interventions	*n* (%)
Diet and lifestyle only	52 (34.2%)
Metformin only	7 (4.6%)
Insulin (basal, bolus, or with metformin)	93 (61.1%)

**Table 3 ijerph-17-07725-t003:** Risk of maternal and neonatal outcomes in women treated for GDM compared to the untreated counterfactual group.

Outcome	GDM Treated *n* = 152	Counterfactual Group ^§^ *n* = 135	Relative Risk (95% CI)
**Maternal outcomes**			
Gestational age at delivery (weeks)	38.2 ± 1.2 *	38.8 ± 1.2	NA
Caesarean sections (%)	32.2 (*n* = 49)	43.0 (*n* = 58)	0.75 (0.56–1.01)
Primary caesarean sections (%)	24.3 (*n* = 33)	30.7 (*n* = 35)	0.79 (0.52–1.18)
Induced delivery	61.8 (*n* = 90)	39.3 (*n* = 53)	1.57 (1.18–1.9)
**Neonatal outcomes**			
Birth weight (g)	3224 ± 499 *	3516 ± 509	NA
Newborn BMI (kg/m^2^)	13.4 ± 1.5 *	14.0 ± 1.4	NA
Large for gestational age (%)	4.6 (*n* = 7)	12.6 (*n* = 17)	0.37 (0.16–0.85)
Small for gestational age (%)	4.6 (*n* = 7)	1.5 (*n* = 2)	3.1 (0.65–14.7)
Neonatal hypoglycaemia (%)	15.8 (*n* = 24)	5.9 (*n* = 8)	2.66 (1.23–5.73)
Admission to special care nursery (%)	19.8 (*n* = 30)	17.0 (*n* = 23)	1.15 (0.71–1.9)
Shoulder dystocia (%)	1.3 (*n* = 2)	0 (*n* = 0)	NA

Values are mean ± standard deviation unless otherwise specified. Relative risk as compared to the counterfactual group. * *p* < 0.05. Non-paired student *t*-test used to compare continuous variables and Mann-Whitney test used to compare non-parametric categorical variables. NA not applicable. ^§^ Counterfactual group selected from women with OGTT just below the diagnostic thresholds, HAPO glucose composite categories ≥9.

**Table 4 ijerph-17-07725-t004:** Risk of maternal and neonatal outcomes in women treated for GDM compared to the untreated counterfactual group, stratified by obesity.

Outcome	GDM BMI ≥ 30 *n* = 79	Counterfactual Group BMI ≥ 30 *n* = 59	Relative Risk 95% CI	GDM BMI < 30 *n* = 73	Counterfactual Group BMI < 30 *n* = 76	Relative Risk 95% CI
Caesarean sections (%)	32.9 (*n* = 26)	55.9 (*n* = 33)	0.59 (0.4–0.87)	31.5 (*n* = 23)	32.9 (*n* = 25)	0.96 (0.6–1.15)
Primary caesarean sections (%)	26.4 (*n* = 19)	45.8 (*n* = 22)	0.58 (0.35–0.94)	20.6 (*n* = 13)	20.3 (*n* = 13)	1.02 (0.51–2.0)
Large for gestational age (%)	5.1 (*n* = 4)	13.6 (*n* = 8)	0.37 (0.12–1.18)	4.1 (*n* = 3)	11.8 (*n* = 9)	0.35 (0.1–1.23)
Small for gestational age (%)	6.3 (*n* = 5)	0 (*n* = 0)	NA	2.7 (*n* = 2)	2.6 (*n* = 2)	1.04 (0.15–7.2)

CI, confidence interval. NA, not applicable.

## Data Availability

The ethics committee has not allowed any sharing of data.

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
