# Peer review of "Estimated Treatment Effects of Tight Glycaemic Targets in Mild Gestational Diabetes Mellitus: A Multiple Cut-Off Regression Discontinuity Study Design"

_ijerph, 2020, doi:10.3390/ijerph17217725_

Round 1

Reviewer 1 Report

In this paper, the authors investigated the treatment effects of tight glycaemic targets in a population universally screened for gestational diabetes mellitus (GDM). They conclude that this treatment reduces LGA outcomes, and caesarean section rates when compared to non-GDM subjects just below the diagnostic thresholds. The paper is interesting and well written. I have only few concerns.

  • The figure 1 is not readable. Please, fix it.
  • In table 2, please, indicate the p value of Neonatal hypoglycaemia.
  • Please, discuss your data in the light of results by Li et al. (Lancet Diabetes Endocrinol 2020; 8: 259­260. DOI: 10.1016/S2213-8587(20)30024-3).

Author Response

Dear reviewers,

Thank you for the helpful comments and suggestions. I have highlighted in red my responses to those comments.

Kind regards,

David Song

 Abstract

  1. Background: place the study in a broad context e.g. Lack of RCT evidence for management of mild hyperglycaemia defined by IADPSG criteria. Describe aim/hypothesis in background e.g. comparison of treatment group to untreated women just below diagnostic thresholds (counterfactual group).

                Added: “As yet there has been no randomized control trials evaluating the effectiveness of treatment of mild GDM diagnosed under the IADPSG/WHO diagnostic thresholds.”

  1. Methods: Describe derivation of regression discontinuity counterfactual group using composite glucose category. Include collection of antenatal and birth data.

Added: “Regression discontinuity study designs estimate treatment effects by comparing outcomes between a treated group to a counterfactual group just below the diagnostic thresholds with the assumption that covariates are similar. The counterfactual group was selected based on composite score based on OGTT plasma glucose categories.. “

  1. Results: Include rates of hypoglycaemia and induction and insulin use as these are mentioned in the conclusion

Added: “This is at the expense of increases in induced deliveries, 61.8% versus 39.3%, relative risk 1.57 (95% CI, 1.18-1.9); notations of neonatal hypoglycaemia, 15.8% versus 5.9%, relative risk 2.66 (95% CI, 1.23-5.73); and high insulin usage 61.1%.”

Introduction

The authors give a clear and concise summary of the study context. Suggestions for additional information are:

  1. As this paper will be part of a special edition for rural communities, it would be valid to introduce diagnosis and management of GDM in a rural context e.g. application of IADPSG criteria selected from a predominantly metropolitan cohort. Access to specialist care putting additional strain on rural clinics with increase in GDM identified. It would also be valid to point out differences in caesarean rates in rural v urban Australia to highlight your point about disparity between settings in lines 54 to 57.

Added: “In rural communities access to specialist care may be limited and the clinical impact of increased GDM prevalence by lowering diagnostic thresholds in these cohorts are yet to be clarified.”

There are no large differences between caesarean section rates between rural and metropolitan centres in Australia. Greater disparities are seen between public and private hospitals.

  1. Your study aims would be clearer if regression discontinuity (RD) is explained first. Suggest moving the description of RD in lines 244 -252 to the introduction to help justify the study design.

Added: “GDM is unusual as there is no natural threshold differentiating disease and non-disease populations. There is a continuous relationship between OGTT plasma glucose levels and clinical outcomes1. We use a multiple cut-off regression discontinuity study design analysing the clinical outcomes around the IADPSG/WHO diagnostic thresholds to estimate treatment effects, utilising the continuous relationship of OGTT plasma glucose to GDM outcomes.”

Methods

The description of the study setting is clear and provides a good background for this regional and rural cohort. Consider clarifying or including the following information:

  1. Please name the local ethics committee. Changed, We received Ballarat Health Services and St John of God Healthcare Human Research Ethics Committee approval for this study (LNR/18/BHSSJOG/16).
  2. This was a retrospective study however line 83 describes the collection of prospective data. Please revise wording.

Removed the word ‘prospective’.

  1. Line 82; please revise to: OGTTs before 24 or after 28 weeks.

Changed

  1. Line 103; describes women diagnosed early in pregnancy however the study inclusion criteria are women with an OGTT between 24 and 28 weeks. Is this sentence required?

Unchanged: for non-GDM women who had multiple OGTTs, we included those done between 24-28 weeks.

  1. Line 115 and 118; consider combining under one sub-heading

Combined headlines, “Subject allocation and HAPO glucose composite score”

  1. Line 138 to 139; This threshold was selected as all GDM subjects, except five, had HAPO glucose composite score of ≥9. The process for selection of the counterfactual should be reported in the methods e.g. The composite score used to define the counterfactual group would select at least 95% of women with GDM. However, arrival at the derived composite score ≥9 should be reported in the results.

Changed as per reviewers comments: Methods: Bandwidth selection for the counterfactual group was selected on the basis of HAPO glucose composite scores where it also selects at least 95% of women with GDM.

  1. Lines 144 to 149; pooling of scores for linear regression: Describe the protocol for pooling e.g. categories with < than x number of women were pooled with the next adjacent category until a minimum of x was obtained. Suggest reporting which categories were pooled as part of the results.

We implied where there were no outcomes for LGA it was pooled an adjacent category, thus N≥1.

Results text

  1. Line 161; remove 3.1 Study population sub-heading.

Changed

  1. Line 165 to 171; this section requires rewriting to match flowchart and could be more concise. Description of the cutpoint of ≥9 should be inserted here.

Rewritten: “Among 1505 women delivering at BHS, 1174 had complete OGTTs (figure 1). There were 162 subjects with OGTT above IADPSG/WHO GDM thresholds. After excluding Overt Diabetes and those who did not attend BHS GDM clinic, there were 152 GDM subjects. Between 24-28 weeks gestation there were 888 subjects after removal of those above diagnostic thresholds and incomplete OGTT. There were 135 subjects with HAPO glucose composite scores ≥9, which are here defined as the counterfactual group as more than 95% of GDM subjects, except 5 (figure 1), have HAPO glucose composite score ≥9.  The counterfactual group represents the highest 15.2% of non-GDM HAPO composite glucose scores.”

  1. Line 188 to 203; highlight significant differences and refer to table – perhaps report P values if * used to designate significance in table but no need to restate results (see Table 2 suggestions).

Changed

  1. Lines 215 to 233; this information is repeated. HAPO glucose categories should either be reported in the methods section or in a supplementary table that can be referred to for the figure and table legends.

HAPO glucose categories moved to the methods section.

Figure 1

This figure is not clear in its current format. A suggested title is: OGTT score and GDM diagnosis in women audited. Suggested changes are:

Changed title as per suggestions.

  1. Remove exclusions box for incomplete OGTT and just report 1174 complete OGTT in 2nd box.

Removed.

  1. First branching should come off OGTT between 24 and 28 weeks not complete OGTT box.

Unchanged as some GDM women were diagnosed before 24-28 weeks.

  1. Report composite scores for GDM as well as non-GDM and report numbers in all score groups not combined group used to define counterfactual group.

Changed

  1. Legend: Define abbreviations and include method to generate composite OGTT score and designate counterfactual group - figure should be stand alone and understandable without reference to the text.

Added legend on how composite OGTT was generated.

Table 1

A suggested title is: Baseline characteristics of women audited, comparing women treated for GDM to the untreated counterfactual group. Suggested changes are:

Changed title as per suggestions

  1. Suggest placing all participants in column one, counterfactual group in column 2 and GDM in column three.

No change: We did not search results for patients with no OGTTs. It would be incomplete.

  1. Use * ** *** to designate significant difference between group.

Changed

  1. List ethnic groups in variable column rather than repeating in group columns.

Changed

  1. Footnote: Define abbreviations e.g. GDM, OGTT, and include method to generate composite OGTT score and designate counterfactual group. Table should be stand alone and understandable without reference to the text.

Added: HAPO glucose composite scores are the sum of HAPO glucose categories at OGTT, fasting, 1-hour, and 2-hour.

The abbreviations GDM, OGTT should be self-explanatory.

Table 2

A suggested title is: Risk of maternal and neonatal outcomes in women treated for GDM compared to the untreated counterfactual group. Suggested changes are:

Changed title as per suggestions.

  1. Reverse order or columns.

Unchanged to maintain consistency with other tables.

  1. Footnote: Define abbreviations and include method to designate counterfactual group. Table should be stand alone and understandable without reference to the text.

Added as per each figure/table.

  1. State that the risk is calculated relative to the counterfactual group.

Changed

  1. Capitalise N for total in group (ie denominator) lower case n used for numerator.

Changed

  1. Suggest use of * ** *** to designate significant difference rather than reporting p value OR report p value for all comparisons.

Changed

  1. Only chi2 p value should be reported. p value from linear regression should be reported in text.

Changed

Table 3

A suggested title is: Risk of maternal and neonatal outcomes in women treated for GDM compared to the untreated counterfactual group, stratified by obesity. Suggested changes are:

Changed title as per suggestions.

  1. Reverse order or columns.

I have kept the order.

  1. Capitalise N for total in group (ie denominator) lower case n used for numerator.

Changed

  1. Footnote: Define abbreviations and include method to designate counterfactual group. Table should be stand alone and understandable without reference to the text.

GDM should be understood in the context of the article/title and the term BMI is universally.

Table 4.

This information should be introduced earlier e.g. after Table 1. and prior to reporting outcomes. Suggested Title: Interventions reported for women diagnosed with GDM

Changed order of tables, table 4 is now table 1.

  1. Insulin row is difficult to read. Suggest listing total with insulin in table and describing basal and bolus amounts in text or reformatting table so that subtype information is clear.

I have simplified the table and added text in results for explanations.

Discussion

As suggested in point 5 above, we suggest moving the description of RD in lines 244 -252 to the introduction to help justify the study design.

Added to the introduction as previously mentioned.

  1. Perhaps introduce your main findings first in context of the study aims e.g. reduction of LGA and CS, but higher induction rates.
  2. Please discuss your data in the light of results by Li et al. (Lancet Diabetes Endocrinol 2020; 8: 259 260. DOI: 10.1016/S2213-8587(20)30024-3) (Reviewer comment)

It is not clear how the study by Li et al would affect the interpretation of our study.

  1. Line 262; suggest reporting GDM for this cohort and the Australian HAPO cohorts as a statistical comparison is not reported in the text, although note that some of the women audited would have been unblinded as per HAPO criteria. Added: Our GDM prevalence of 13.5% was similar to Australian HAPO centres, 12.4 to 15.3%

Would also be worthwhile comparing OGTT values to those in the ACHOIS RCT perhaps when discussing the different treatment targets used.

  1. Line 269; is not complete. Removed, it was a subtitle originally.
  2. Line 327; change to: RCTs select.

Changed

  1. Line 333; Does not read well – They would have been.

Changed, it was a grammatical mistake.  “We would expect higher baseline adverse outcomes…”

  1. Line 343; change to: Of women with GDM…

Changed

  1. Line 342 to 348; move earlier in discussion prior to discussion of outcomes.

Moved to earlier in results and discussions.

  1. Line 361; suggest removing statement regarding no recollection of severe hypoglycaemia.

Removed

Minor points/typos

  1. Line 72; remove capitals; oral glucose tolerance test.

Changed

  1. Line 85; please define BMI and calculation.

Written in full, Body mass index, added units to denote calculation kg/m2

  1. Use consistent terminology for OGTT samples e.g. fasting v fasting glucose v fasting plasma glucose, 1-hour v 1 hour v 1-h etc.

Changed

  1. Be consistent with spacing between symbols and units in text and tables e.g. ≥ _10.0 v ≥10.0; ± 1.1 v ±1.1; kg/m2 v kgm2
  2. Line 167; sentence begins with And.

Removed ‘And’

  1. Be consistent with units e.g ethnicity % is only used for Caucasian ethnicity and not others

Changed table format for clarity

Editors additional comments regarding composite OGTT score

In your response to the editors query regarding the use of a composite score you mentioned that there is no evidence for association between composite OGTT z-score and adverse birth outcomes. The HAPO Study Cooperative Research Group have published data that shows the association between composite OGTT z-score and adverse outcomes is predominantly stronger compared to 1 SD shift in HbA1c and individual OGTT values.(1)

In another paper the HAPO Study Cooperative Research Group calculated a composite OGTT z-score and categorised the z-score sum as normal (<=0.539, no GDM (61.8%)) intermediate (>0.539, no GDM (22.2%)) and GDM (16.1%)).(2) The cut point was selected to have approximately the same proportions as derived by stratification by BMI. In this paper the normal group was used as the reference group to compare to the GDM group, showing a 339g difference in birthweight for babies of obese GDM v normal/underweight women classified as normal.

Your generation of a counterfactual group, just below the current diagnostic thresholds for GDM by glucose category is a valid comparison group for the GDM group who have received management. One potential issue with using glucose category to create an OGTT composite score is that categorisation, particularly FPG categorisation, is affected by preanalytical process, which you do acknowledge in the discussion. The OCC cohort where fluoride-citrate tubes were used is a good example of where the HAPO FPG categories are not applicable.(3) In this cohort 46% of participants fell in category 5 or higher but high incidence of LGA (>10%) was not observed until category 7.

A composite z-score would be calculated using local mean and sd. In terms of use of FC mix tubes, whilst this would eliminate some of the testing variability, there will always be analytical error. Combined with the fact that there is a continuous linear association between glucose and adverse outcomes, regardless of preanalytical protocol, indicates that women with similar z-scores but just below the diagnostic thresholds will have similar or only slightly lower risk to those just above the cut-points and validates your regression discontinuity approach.

As the composite z-score has already been validated and published by the HAPO Study Cooperative Group this may be a more accepted method to generate a composite OGTT score. Our recommendations would be to repeat the analysis using composite z-score and selecting the threshold for the counterfactual group as the minimum composite z-score for 95% of the GDM group. At the very least it would be interesting to determine how much overlap there is between the two counterfactual groups (categorical composite v z-score composite).

  1. Lowe LP, Metzger BE, Dyer AR, et al. Hyperglycemia and Adverse Pregnancy Outcome (HAPO) Study: associations of maternal A1C and glucose with pregnancy outcomes. Diabetes Care. 2012;35(3):574‐580. doi:10.2337/dc11-1687

  1. Catalano PM, McIntyre HD, Cruickshank JK, et al. The hyperglycemia and adverse pregnancy outcome study: associations of GDM and obesity with pregnancy outcomes. Diabetes Care. 2012;35(4):780‐786. doi:10.2337/dc11-1790

  1. McIntyre H.D., Jensen D.M., Jensen R.C., et al. Gestational Diabetes Mellitus: Does One Size Fit All? A Challenge to Uniform Worldwide Diagnostic Thresholds. Diabetes Care. 2018; 41(7): 1339-1342.

Dear reviewers,

Thank you for the helpful comments and suggestions. I have highlighted in red my responses to those comments.

Kind regards,

David Song

 Abstract

  1. Background: place the study in a broad context e.g. Lack of RCT evidence for management of mild hyperglycaemia defined by IADPSG criteria. Describe aim/hypothesis in background e.g. comparison of treatment group to untreated women just below diagnostic thresholds (counterfactual group).

                Added: “As yet there has been no randomized control trials evaluating the effectiveness of treatment of mild GDM diagnosed under the IADPSG/WHO diagnostic thresholds.”

  1. Methods: Describe derivation of regression discontinuity counterfactual group using composite glucose category. Include collection of antenatal and birth data.

Added: “Regression discontinuity study designs estimate treatment effects by comparing outcomes between a treated group to a counterfactual group just below the diagnostic thresholds with the assumption that covariates are similar. The counterfactual group was selected based on composite score based on OGTT plasma glucose categories.. “

  1. Results: Include rates of hypoglycaemia and induction and insulin use as these are mentioned in the conclusion

Added: “This is at the expense of increases in induced deliveries, 61.8% versus 39.3%, relative risk 1.57 (95% CI, 1.18-1.9); notations of neonatal hypoglycaemia, 15.8% versus 5.9%, relative risk 2.66 (95% CI, 1.23-5.73); and high insulin usage 61.1%.”

Introduction

The authors give a clear and concise summary of the study context. Suggestions for additional information are:

  1. As this paper will be part of a special edition for rural communities, it would be valid to introduce diagnosis and management of GDM in a rural context e.g. application of IADPSG criteria selected from a predominantly metropolitan cohort. Access to specialist care putting additional strain on rural clinics with increase in GDM identified. It would also be valid to point out differences in caesarean rates in rural v urban Australia to highlight your point about disparity between settings in lines 54 to 57.

Added: “In rural communities access to specialist care may be limited and the clinical impact of increased GDM prevalence by lowering diagnostic thresholds in these cohorts are yet to be clarified.”

There are no large differences between caesarean section rates between rural and metropolitan centres in Australia. Greater disparities are seen between public and private hospitals.

  1. Your study aims would be clearer if regression discontinuity (RD) is explained first. Suggest moving the description of RD in lines 244 -252 to the introduction to help justify the study design.

Added: “GDM is unusual as there is no natural threshold differentiating disease and non-disease populations. There is a continuous relationship between OGTT plasma glucose levels and clinical outcomes1. We use a multiple cut-off regression discontinuity study design analysing the clinical outcomes around the IADPSG/WHO diagnostic thresholds to estimate treatment effects, utilising the continuous relationship of OGTT plasma glucose to GDM outcomes.”

Methods

The description of the study setting is clear and provides a good background for this regional and rural cohort. Consider clarifying or including the following information:

  1. Please name the local ethics committee. Changed, We received Ballarat Health Services and St John of God Healthcare Human Research Ethics Committee approval for this study (LNR/18/BHSSJOG/16).
  2. This was a retrospective study however line 83 describes the collection of prospective data. Please revise wording.

Removed the word ‘prospective’.

  1. Line 82; please revise to: OGTTs before 24 or after 28 weeks.

Changed

  1. Line 103; describes women diagnosed early in pregnancy however the study inclusion criteria are women with an OGTT between 24 and 28 weeks. Is this sentence required?

Unchanged: for non-GDM women who had multiple OGTTs, we included those done between 24-28 weeks.

  1. Line 115 and 118; consider combining under one sub-heading

Combined headlines, “Subject allocation and HAPO glucose composite score”

  1. Line 138 to 139; This threshold was selected as all GDM subjects, except five, had HAPO glucose composite score of ≥9. The process for selection of the counterfactual should be reported in the methods e.g. The composite score used to define the counterfactual group would select at least 95% of women with GDM. However, arrival at the derived composite score ≥9 should be reported in the results.

Changed as per reviewers comments: Methods: Bandwidth selection for the counterfactual group was selected on the basis of HAPO glucose composite scores where it also selects at least 95% of women with GDM.

  1. Lines 144 to 149; pooling of scores for linear regression: Describe the protocol for pooling e.g. categories with < than x number of women were pooled with the next adjacent category until a minimum of x was obtained. Suggest reporting which categories were pooled as part of the results.

We implied where there were no outcomes for LGA it was pooled an adjacent category, thus N≥1.

Results text

  1. Line 161; remove 3.1 Study population sub-heading.

Changed

  1. Line 165 to 171; this section requires rewriting to match flowchart and could be more concise. Description of the cutpoint of ≥9 should be inserted here.

Rewritten: “Among 1505 women delivering at BHS, 1174 had complete OGTTs (figure 1). There were 162 subjects with OGTT above IADPSG/WHO GDM thresholds. After excluding Overt Diabetes and those who did not attend BHS GDM clinic, there were 152 GDM subjects. Between 24-28 weeks gestation there were 888 subjects after removal of those above diagnostic thresholds and incomplete OGTT. There were 135 subjects with HAPO glucose composite scores ≥9, which are here defined as the counterfactual group as more than 95% of GDM subjects, except 5 (figure 1), have HAPO glucose composite score ≥9.  The counterfactual group represents the highest 15.2% of non-GDM HAPO composite glucose scores.”

  1. Line 188 to 203; highlight significant differences and refer to table – perhaps report P values if * used to designate significance in table but no need to restate results (see Table 2 suggestions).

Changed

  1. Lines 215 to 233; this information is repeated. HAPO glucose categories should either be reported in the methods section or in a supplementary table that can be referred to for the figure and table legends.

HAPO glucose categories moved to the methods section.

Figure 1

This figure is not clear in its current format. A suggested title is: OGTT score and GDM diagnosis in women audited. Suggested changes are:

Changed title as per suggestions.

  1. Remove exclusions box for incomplete OGTT and just report 1174 complete OGTT in 2nd box.

Removed.

  1. First branching should come off OGTT between 24 and 28 weeks not complete OGTT box.

Unchanged as some GDM women were diagnosed before 24-28 weeks.

  1. Report composite scores for GDM as well as non-GDM and report numbers in all score groups not combined group used to define counterfactual group.

Changed

  1. Legend: Define abbreviations and include method to generate composite OGTT score and designate counterfactual group - figure should be stand alone and understandable without reference to the text.

Added legend on how composite OGTT was generated.

Table 1

A suggested title is: Baseline characteristics of women audited, comparing women treated for GDM to the untreated counterfactual group. Suggested changes are:

Changed title as per suggestions

  1. Suggest placing all participants in column one, counterfactual group in column 2 and GDM in column three.

No change: We did not search results for patients with no OGTTs. It would be incomplete.

  1. Use * ** *** to designate significant difference between group.

Changed

  1. List ethnic groups in variable column rather than repeating in group columns.

Changed

  1. Footnote: Define abbreviations e.g. GDM, OGTT, and include method to generate composite OGTT score and designate counterfactual group. Table should be stand alone and understandable without reference to the text.

Added: HAPO glucose composite scores are the sum of HAPO glucose categories at OGTT, fasting, 1-hour, and 2-hour.

The abbreviations GDM, OGTT should be self-explanatory.

Table 2

A suggested title is: Risk of maternal and neonatal outcomes in women treated for GDM compared to the untreated counterfactual group. Suggested changes are:

Changed title as per suggestions.

  1. Reverse order or columns.

Unchanged to maintain consistency with other tables.

  1. Footnote: Define abbreviations and include method to designate counterfactual group. Table should be stand alone and understandable without reference to the text.

Added as per each figure/table.

  1. State that the risk is calculated relative to the counterfactual group.

Changed

  1. Capitalise N for total in group (ie denominator) lower case n used for numerator.

Changed

  1. Suggest use of * ** *** to designate significant difference rather than reporting p value OR report p value for all comparisons.

Changed

  1. Only chi2 p value should be reported. p value from linear regression should be reported in text.

Changed

Table 3

A suggested title is: Risk of maternal and neonatal outcomes in women treated for GDM compared to the untreated counterfactual group, stratified by obesity. Suggested changes are:

Changed title as per suggestions.

  1. Reverse order or columns.

I have kept the order.

  1. Capitalise N for total in group (ie denominator) lower case n used for numerator.

Changed

  1. Footnote: Define abbreviations and include method to designate counterfactual group. Table should be stand alone and understandable without reference to the text.

GDM should be understood in the context of the article/title and the term BMI is universally.

Table 4.

This information should be introduced earlier e.g. after Table 1. and prior to reporting outcomes. Suggested Title: Interventions reported for women diagnosed with GDM

Changed order of tables, table 4 is now table 1.

  1. Insulin row is difficult to read. Suggest listing total with insulin in table and describing basal and bolus amounts in text or reformatting table so that subtype information is clear.

I have simplified the table and added text in results for explanations.

Discussion

As suggested in point 5 above, we suggest moving the description of RD in lines 244 -252 to the introduction to help justify the study design.

Added to the introduction as previously mentioned.

  1. Perhaps introduce your main findings first in context of the study aims e.g. reduction of LGA and CS, but higher induction rates.
  2. Please discuss your data in the light of results by Li et al. (Lancet Diabetes Endocrinol 2020; 8: 259 260. DOI: 10.1016/S2213-8587(20)30024-3) (Reviewer comment)

It is not clear how the study by Li et al would affect the interpretation of our study.

  1. Line 262; suggest reporting GDM for this cohort and the Australian HAPO cohorts as a statistical comparison is not reported in the text, although note that some of the women audited would have been unblinded as per HAPO criteria. Added: Our GDM prevalence of 13.5% was similar to Australian HAPO centres, 12.4 to 15.3%

Would also be worthwhile comparing OGTT values to those in the ACHOIS RCT perhaps when discussing the different treatment targets used.

  1. Line 269; is not complete. Removed, it was a subtitle originally.
  2. Line 327; change to: RCTs select.

Changed

  1. Line 333; Does not read well – They would have been.

Changed, it was a grammatical mistake.  “We would expect higher baseline adverse outcomes…”

  1. Line 343; change to: Of women with GDM…

Changed

  1. Line 342 to 348; move earlier in discussion prior to discussion of outcomes.

Moved to earlier in results and discussions.

  1. Line 361; suggest removing statement regarding no recollection of severe hypoglycaemia.

Removed

Minor points/typos

  1. Line 72; remove capitals; oral glucose tolerance test.

Changed

  1. Line 85; please define BMI and calculation.

Written in full, Body mass index, added units to denote calculation kg/m2

  1. Use consistent terminology for OGTT samples e.g. fasting v fasting glucose v fasting plasma glucose, 1-hour v 1 hour v 1-h etc.

Changed

  1. Be consistent with spacing between symbols and units in text and tables e.g. ≥ _10.0 v ≥10.0; ± 1.1 v ±1.1; kg/m2 v kgm2
  2. Line 167; sentence begins with And.

Removed ‘And’

  1. Be consistent with units e.g ethnicity % is only used for Caucasian ethnicity and not others

Changed table format for clarity

Editors additional comments regarding composite OGTT score

In your response to the editors query regarding the use of a composite score you mentioned that there is no evidence for association between composite OGTT z-score and adverse birth outcomes. The HAPO Study Cooperative Research Group have published data that shows the association between composite OGTT z-score and adverse outcomes is predominantly stronger compared to 1 SD shift in HbA1c and individual OGTT values.(1)

In another paper the HAPO Study Cooperative Research Group calculated a composite OGTT z-score and categorised the z-score sum as normal (<=0.539, no GDM (61.8%)) intermediate (>0.539, no GDM (22.2%)) and GDM (16.1%)).(2) The cut point was selected to have approximately the same proportions as derived by stratification by BMI. In this paper the normal group was used as the reference group to compare to the GDM group, showing a 339g difference in birthweight for babies of obese GDM v normal/underweight women classified as normal.

Your generation of a counterfactual group, just below the current diagnostic thresholds for GDM by glucose category is a valid comparison group for the GDM group who have received management. One potential issue with using glucose category to create an OGTT composite score is that categorisation, particularly FPG categorisation, is affected by preanalytical process, which you do acknowledge in the discussion. The OCC cohort where fluoride-citrate tubes were used is a good example of where the HAPO FPG categories are not applicable.(3) In this cohort 46% of participants fell in category 5 or higher but high incidence of LGA (>10%) was not observed until category 7.

A composite z-score would be calculated using local mean and sd. In terms of use of FC mix tubes, whilst this would eliminate some of the testing variability, there will always be analytical error. Combined with the fact that there is a continuous linear association between glucose and adverse outcomes, regardless of preanalytical protocol, indicates that women with similar z-scores but just below the diagnostic thresholds will have similar or only slightly lower risk to those just above the cut-points and validates your regression discontinuity approach.

As the composite z-score has already been validated and published by the HAPO Study Cooperative Group this may be a more accepted method to generate a composite OGTT score. Our recommendations would be to repeat the analysis using composite z-score and selecting the threshold for the counterfactual group as the minimum composite z-score for 95% of the GDM group. At the very least it would be interesting to determine how much overlap there is between the two counterfactual groups (categorical composite v z-score composite).

  1. Lowe LP, Metzger BE, Dyer AR, et al. Hyperglycemia and Adverse Pregnancy Outcome (HAPO) Study: associations of maternal A1C and glucose with pregnancy outcomes. Diabetes Care. 2012;35(3):574‐580. doi:10.2337/dc11-1687

  1. Catalano PM, McIntyre HD, Cruickshank JK, et al. The hyperglycemia and adverse pregnancy outcome study: associations of GDM and obesity with pregnancy outcomes. Diabetes Care. 2012;35(4):780‐786. doi:10.2337/dc11-1790

  1. McIntyre H.D., Jensen D.M., Jensen R.C., et al. Gestational Diabetes Mellitus: Does One Size Fit All? A Challenge to Uniform Worldwide Diagnostic Thresholds. Diabetes Care. 2018; 41(7): 1339-1342.

Reviewer 2 Report

These are my general/specific comments.

In their paper entitled “Estimated treatment effects of tight glycaemic targets in mild 
gestational diabetes mellitus: a multiple cut-off regression discontinuity 
study design”, the authors investigated the treatment effects of tight glycaemic targets in a population universally screened according to the International Association of Diabetes and Pregnant Study Groups/World Health Organisation gestational diabetes mellitus (GDM) guidelines.

Comments:

  1. The authors should review the English throughout the paper.

  1. In their Abstract/Introduction it is recommended that the authors state an hypothesis.

  1. In the results section, the authors should describe in more detail all the results they got. The authors should also considerer to analyze these data using machine learning in order to clarify which component is on the basis of the differences they obtained.

  1. The authors should format figure 1.

  1. In the discussion section authors should consider discussing in more detail the mechanism involved in the tight glycaemic targets reduced rates of large for gestational age and caesarean sections.

Author Response

Dear reviewers,

Thank you for the helpful comments and suggestions. I have highlighted in red my responses to those comments.

Kind regards,

David Song

 Abstract

  1. Background: place the study in a broad context e.g. Lack of RCT evidence for management of mild hyperglycaemia defined by IADPSG criteria. Describe aim/hypothesis in background e.g. comparison of treatment group to untreated women just below diagnostic thresholds (counterfactual group).

                Added: “As yet there has been no randomized control trials evaluating the effectiveness of treatment of mild GDM diagnosed under the IADPSG/WHO diagnostic thresholds.”

  1. Methods: Describe derivation of regression discontinuity counterfactual group using composite glucose category. Include collection of antenatal and birth data.

Added: “Regression discontinuity study designs estimate treatment effects by comparing outcomes between a treated group to a counterfactual group just below the diagnostic thresholds with the assumption that covariates are similar. The counterfactual group was selected based on composite score based on OGTT plasma glucose categories.. “

  1. Results: Include rates of hypoglycaemia and induction and insulin use as these are mentioned in the conclusion

Added: “This is at the expense of increases in induced deliveries, 61.8% versus 39.3%, relative risk 1.57 (95% CI, 1.18-1.9); notations of neonatal hypoglycaemia, 15.8% versus 5.9%, relative risk 2.66 (95% CI, 1.23-5.73); and high insulin usage 61.1%.”

Introduction

The authors give a clear and concise summary of the study context. Suggestions for additional information are:

  1. As this paper will be part of a special edition for rural communities, it would be valid to introduce diagnosis and management of GDM in a rural context e.g. application of IADPSG criteria selected from a predominantly metropolitan cohort. Access to specialist care putting additional strain on rural clinics with increase in GDM identified. It would also be valid to point out differences in caesarean rates in rural v urban Australia to highlight your point about disparity between settings in lines 54 to 57.

Added: “In rural communities access to specialist care may be limited and the clinical impact of increased GDM prevalence by lowering diagnostic thresholds in these cohorts are yet to be clarified.”

There are no large differences between caesarean section rates between rural and metropolitan centres in Australia. Greater disparities are seen between public and private hospitals.

  1. Your study aims would be clearer if regression discontinuity (RD) is explained first. Suggest moving the description of RD in lines 244 -252 to the introduction to help justify the study design.

Added: “GDM is unusual as there is no natural threshold differentiating disease and non-disease populations. There is a continuous relationship between OGTT plasma glucose levels and clinical outcomes1. We use a multiple cut-off regression discontinuity study design analysing the clinical outcomes around the IADPSG/WHO diagnostic thresholds to estimate treatment effects, utilising the continuous relationship of OGTT plasma glucose to GDM outcomes.”

Methods

The description of the study setting is clear and provides a good background for this regional and rural cohort. Consider clarifying or including the following information:

  1. Please name the local ethics committee. Changed, We received Ballarat Health Services and St John of God Healthcare Human Research Ethics Committee approval for this study (LNR/18/BHSSJOG/16).
  2. This was a retrospective study however line 83 describes the collection of prospective data. Please revise wording.

Removed the word ‘prospective’.

  1. Line 82; please revise to: OGTTs before 24 or after 28 weeks.

Changed

  1. Line 103; describes women diagnosed early in pregnancy however the study inclusion criteria are women with an OGTT between 24 and 28 weeks. Is this sentence required?

Unchanged: for non-GDM women who had multiple OGTTs, we included those done between 24-28 weeks.

  1. Line 115 and 118; consider combining under one sub-heading

Combined headlines, “Subject allocation and HAPO glucose composite score”

  1. Line 138 to 139; This threshold was selected as all GDM subjects, except five, had HAPO glucose composite score of ≥9. The process for selection of the counterfactual should be reported in the methods e.g. The composite score used to define the counterfactual group would select at least 95% of women with GDM. However, arrival at the derived composite score ≥9 should be reported in the results.

Changed as per reviewers comments: Methods: Bandwidth selection for the counterfactual group was selected on the basis of HAPO glucose composite scores where it also selects at least 95% of women with GDM.

  1. Lines 144 to 149; pooling of scores for linear regression: Describe the protocol for pooling e.g. categories with < than x number of women were pooled with the next adjacent category until a minimum of x was obtained. Suggest reporting which categories were pooled as part of the results.

We implied where there were no outcomes for LGA it was pooled an adjacent category, thus N≥1.

Results text

  1. Line 161; remove 3.1 Study population sub-heading.

Changed

  1. Line 165 to 171; this section requires rewriting to match flowchart and could be more concise. Description of the cutpoint of ≥9 should be inserted here.

Rewritten: “Among 1505 women delivering at BHS, 1174 had complete OGTTs (figure 1). There were 162 subjects with OGTT above IADPSG/WHO GDM thresholds. After excluding Overt Diabetes and those who did not attend BHS GDM clinic, there were 152 GDM subjects. Between 24-28 weeks gestation there were 888 subjects after removal of those above diagnostic thresholds and incomplete OGTT. There were 135 subjects with HAPO glucose composite scores ≥9, which are here defined as the counterfactual group as more than 95% of GDM subjects, except 5 (figure 1), have HAPO glucose composite score ≥9.  The counterfactual group represents the highest 15.2% of non-GDM HAPO composite glucose scores.”

  1. Line 188 to 203; highlight significant differences and refer to table – perhaps report P values if * used to designate significance in table but no need to restate results (see Table 2 suggestions).

Changed

  1. Lines 215 to 233; this information is repeated. HAPO glucose categories should either be reported in the methods section or in a supplementary table that can be referred to for the figure and table legends.

HAPO glucose categories moved to the methods section.

Figure 1

This figure is not clear in its current format. A suggested title is: OGTT score and GDM diagnosis in women audited. Suggested changes are:

Changed title as per suggestions.

  1. Remove exclusions box for incomplete OGTT and just report 1174 complete OGTT in 2nd box.

Removed.

  1. First branching should come off OGTT between 24 and 28 weeks not complete OGTT box.

Unchanged as some GDM women were diagnosed before 24-28 weeks.

  1. Report composite scores for GDM as well as non-GDM and report numbers in all score groups not combined group used to define counterfactual group.

Changed

  1. Legend: Define abbreviations and include method to generate composite OGTT score and designate counterfactual group - figure should be stand alone and understandable without reference to the text.

Added legend on how composite OGTT was generated.

Table 1

A suggested title is: Baseline characteristics of women audited, comparing women treated for GDM to the untreated counterfactual group. Suggested changes are:

Changed title as per suggestions

  1. Suggest placing all participants in column one, counterfactual group in column 2 and GDM in column three.

No change: We did not search results for patients with no OGTTs. It would be incomplete.

  1. Use * ** *** to designate significant difference between group.

Changed

  1. List ethnic groups in variable column rather than repeating in group columns.

Changed

  1. Footnote: Define abbreviations e.g. GDM, OGTT, and include method to generate composite OGTT score and designate counterfactual group. Table should be stand alone and understandable without reference to the text.

Added: HAPO glucose composite scores are the sum of HAPO glucose categories at OGTT, fasting, 1-hour, and 2-hour.

The abbreviations GDM, OGTT should be self-explanatory.

Table 2

A suggested title is: Risk of maternal and neonatal outcomes in women treated for GDM compared to the untreated counterfactual group. Suggested changes are:

Changed title as per suggestions.

  1. Reverse order or columns.

Unchanged to maintain consistency with other tables.

  1. Footnote: Define abbreviations and include method to designate counterfactual group. Table should be stand alone and understandable without reference to the text.

Added as per each figure/table.

  1. State that the risk is calculated relative to the counterfactual group.

Changed

  1. Capitalise N for total in group (ie denominator) lower case n used for numerator.

Changed

  1. Suggest use of * ** *** to designate significant difference rather than reporting p value OR report p value for all comparisons.

Changed

  1. Only chi2 p value should be reported. p value from linear regression should be reported in text.

Changed

Table 3

A suggested title is: Risk of maternal and neonatal outcomes in women treated for GDM compared to the untreated counterfactual group, stratified by obesity. Suggested changes are:

Changed title as per suggestions.

  1. Reverse order or columns.

I have kept the order.

  1. Capitalise N for total in group (ie denominator) lower case n used for numerator.

Changed

  1. Footnote: Define abbreviations and include method to designate counterfactual group. Table should be stand alone and understandable without reference to the text.

GDM should be understood in the context of the article/title and the term BMI is universally.

Table 4.

This information should be introduced earlier e.g. after Table 1. and prior to reporting outcomes. Suggested Title: Interventions reported for women diagnosed with GDM

Changed order of tables, table 4 is now table 1.

  1. Insulin row is difficult to read. Suggest listing total with insulin in table and describing basal and bolus amounts in text or reformatting table so that subtype information is clear.

I have simplified the table and added text in results for explanations.

Discussion

As suggested in point 5 above, we suggest moving the description of RD in lines 244 -252 to the introduction to help justify the study design.

Added to the introduction as previously mentioned.

  1. Perhaps introduce your main findings first in context of the study aims e.g. reduction of LGA and CS, but higher induction rates.
  2. Please discuss your data in the light of results by Li et al. (Lancet Diabetes Endocrinol 2020; 8: 259 260. DOI: 10.1016/S2213-8587(20)30024-3) (Reviewer comment)

It is not clear how the study by Li et al would affect the interpretation of our study.

  1. Line 262; suggest reporting GDM for this cohort and the Australian HAPO cohorts as a statistical comparison is not reported in the text, although note that some of the women audited would have been unblinded as per HAPO criteria. Added: Our GDM prevalence of 13.5% was similar to Australian HAPO centres, 12.4 to 15.3%

Would also be worthwhile comparing OGTT values to those in the ACHOIS RCT perhaps when discussing the different treatment targets used.

  1. Line 269; is not complete. Removed, it was a subtitle originally.
  2. Line 327; change to: RCTs select.

Changed

  1. Line 333; Does not read well – They would have been.

Changed, it was a grammatical mistake.  “We would expect higher baseline adverse outcomes…”

  1. Line 343; change to: Of women with GDM…

Changed

  1. Line 342 to 348; move earlier in discussion prior to discussion of outcomes.

Moved to earlier in results and discussions.

  1. Line 361; suggest removing statement regarding no recollection of severe hypoglycaemia.

Removed

Minor points/typos

  1. Line 72; remove capitals; oral glucose tolerance test.

Changed

  1. Line 85; please define BMI and calculation.

Written in full, Body mass index, added units to denote calculation kg/m2

  1. Use consistent terminology for OGTT samples e.g. fasting v fasting glucose v fasting plasma glucose, 1-hour v 1 hour v 1-h etc.

Changed

  1. Be consistent with spacing between symbols and units in text and tables e.g. ≥ _10.0 v ≥10.0; ± 1.1 v ±1.1; kg/m2 v kgm2
  2. Line 167; sentence begins with And.

Removed ‘And’

  1. Be consistent with units e.g ethnicity % is only used for Caucasian ethnicity and not others

Changed table format for clarity

Editors additional comments regarding composite OGTT score

In your response to the editors query regarding the use of a composite score you mentioned that there is no evidence for association between composite OGTT z-score and adverse birth outcomes. The HAPO Study Cooperative Research Group have published data that shows the association between composite OGTT z-score and adverse outcomes is predominantly stronger compared to 1 SD shift in HbA1c and individual OGTT values.(1)

In another paper the HAPO Study Cooperative Research Group calculated a composite OGTT z-score and categorised the z-score sum as normal (<=0.539, no GDM (61.8%)) intermediate (>0.539, no GDM (22.2%)) and GDM (16.1%)).(2) The cut point was selected to have approximately the same proportions as derived by stratification by BMI. In this paper the normal group was used as the reference group to compare to the GDM group, showing a 339g difference in birthweight for babies of obese GDM v normal/underweight women classified as normal.

Your generation of a counterfactual group, just below the current diagnostic thresholds for GDM by glucose category is a valid comparison group for the GDM group who have received management. One potential issue with using glucose category to create an OGTT composite score is that categorisation, particularly FPG categorisation, is affected by preanalytical process, which you do acknowledge in the discussion. The OCC cohort where fluoride-citrate tubes were used is a good example of where the HAPO FPG categories are not applicable.(3) In this cohort 46% of participants fell in category 5 or higher but high incidence of LGA (>10%) was not observed until category 7.

A composite z-score would be calculated using local mean and sd. In terms of use of FC mix tubes, whilst this would eliminate some of the testing variability, there will always be analytical error. Combined with the fact that there is a continuous linear association between glucose and adverse outcomes, regardless of preanalytical protocol, indicates that women with similar z-scores but just below the diagnostic thresholds will have similar or only slightly lower risk to those just above the cut-points and validates your regression discontinuity approach.

As the composite z-score has already been validated and published by the HAPO Study Cooperative Group this may be a more accepted method to generate a composite OGTT score. Our recommendations would be to repeat the analysis using composite z-score and selecting the threshold for the counterfactual group as the minimum composite z-score for 95% of the GDM group. At the very least it would be interesting to determine how much overlap there is between the two counterfactual groups (categorical composite v z-score composite).

  1. Lowe LP, Metzger BE, Dyer AR, et al. Hyperglycemia and Adverse Pregnancy Outcome (HAPO) Study: associations of maternal A1C and glucose with pregnancy outcomes. Diabetes Care. 2012;35(3):574‐580. doi:10.2337/dc11-1687

  1. Catalano PM, McIntyre HD, Cruickshank JK, et al. The hyperglycemia and adverse pregnancy outcome study: associations of GDM and obesity with pregnancy outcomes. Diabetes Care. 2012;35(4):780‐786. doi:10.2337/dc11-1790

  1. McIntyre H.D., Jensen D.M., Jensen R.C., et al. Gestational Diabetes Mellitus: Does One Size Fit All? A Challenge to Uniform Worldwide Diagnostic Thresholds. Diabetes Care. 2018; 41(7): 1339-1342.

Reviewer 3 Report

This is a well written interesting manuscript employing appropriate methodology that will contribute to the literature.

Author Response

(The authors gave the same response as above.)

Reviewer 4 Report

The manuscript, which title is Estimated treatment effects of tight glycaemic targets 2 in mild gestational diabetes mellitus: a multiple cut off regression discontinuity study design, is very interesting. The authors should renew the Figure 1. Subject selection and allocation and all tables. These results could not support their hypothesis.

Author Response

(The authors gave the same response as above.)

Reviewer 5 Report

The study by Song et al investigates the treatment effects of tight glycaemic targets in a retrospective observational cohort that was screened for gestational diabetes mellitus (GDM). The study design is appropriate and the conclusions are in line with the observations. My minor comments can be found below: 

  1. Figure 1: Please correct the overlapping text boxes. The content is not legible
  2. The authors should discuss the possible reasons behind the observation for neo-natal hypoglycaemia in the treated v/s the counterfactual group. Were the babies with neo-natal hypoglycaemia followed up after birth to check for long term consequences if any on their ability to regulate their blood sugar levels?

Author Response

(The authors gave the same response as above.)

Round 2

Reviewer 2 Report

These are my general/specific comments.

In revision of the paper entitled “Estimated treatment effects of tight glycaemic targets in mild gestational diabetes mellitus: a multiple cut-off regression discontinuity 
study design”, the authors investigated the treatment effects of tight glycaemic targets in a population universally screened according to the International Association of Diabetes and Pregnant Study Groups/World Health Organisation gestational diabetes mellitus (GDM) guidelines.

Comments:

  1. It was suggested to state an hypothesis in the Abstract/Introduction and the authors did not include it.

  1. The authors improved the results section.

  1. In the discussion section the authors should discuss in more detail the mechanism involved in the tight glycaemic targets reduced rates of large for gestational age and caesarean sections.

Author Response

Dear editors,

Thanks for reviewing my manuscript and for those helpful comments. I have made alterations to the manuscript in light of the second round of comments. In the manuscript, we highlighted the changes in green. See below for responses to the reviewer's comments.

Kind regards,

David Song

It was suggested to state an hypothesis in the Abstract/Introduction and the authors did not include it.

            I have included in the abstract, “We hypothesize that tight glycaemic control in pregnant women diagnosed with GDM will result in similar clinical outcomes to women just below the diagnostic thresholds.”

The authors improved the results section.

In the discussion section the authors should discuss in more detail the mechanism involved in the tight glycaemic targets reduced rates of large for gestational age and caesarean sections.

The following was added in the Discussion:

4.4. Clinical outcomes

Treated GDM patients had a relative risk of LGA of 0.37, reducing the absolute risk from 12.6 to 4.6%. The RCTs, ACHOIS20 and MFMU 21, reported a decrease in LGA rates from 22 to 13%, 13% to 7.1%, respectively. As our CBG targets were lower it was not surprising that our rates of foetal overgrowth were also lower. The Pedersen hypothesis that states that maternal hyperglycaemia results in foetal hyperinsulinemia, glucose utilization, and subsequent fetal adiposity24. The HAPO study demonstrated the continuous linear associations between maternal glycaemia and cord c-peptide levels and rates of LGA13. It follows that lowering maternal glycaemia should lower rates of LGA and tight maternal glycaemic control should result in lower rates of LGA. Our study further supports the Pedersen hypothesis by demonstrating that tight glycaemic targets lower rates of LGA, even when compared to patients just below the diagnostic thresholds.

The HAPO study demonstrated that both maternal BMI and glycaemia have strong independent associations with caesarean section rates28. However, the HAPO study could not conclude that there was a causal relationship between increased maternal BMI and caesarean sections rates28. Our subgroup analysis of patients based on BMI may suggest tight glycaemic targets in women with BMI <30 kg/m2 does not reduce the risk of caesarean sections. As caesarean section rates decreased in women with GDM with BMI ≥30 kg/m2, it is obesity coupled with maternal glycaemia that are possibly the important factors in determining rates of caesarean sections.

Reviewer 4 Report

The aim of manuscript is interesting. However, there are several major questions in the manuscript. First, the flow-chart (figure 1) is difficult to comprehension, such as the provided the reason and evidence of categories. More, the author must illustrate the means of all arrow in the figure 1. Secondary, the authors should provide the process of grouping to GDM and counterfactual groups in the methods. Third, the authors should provide the reason or evidence of adjustment in table 1. Fourth, it is difficult to support their hypothesis based on that the results have no adjustment.

Author Response

Dear editors,

Thanks for reviewing my manuscript and for those helpful comments. I have made alterations to the manuscript in light of the second round of comments. In the manuscript we highlighted the changes in green.

Kind regards,

David Song

It was suggested to state an hypothesis in the Abstract/Introduction and the authors did not include it.

            I have included in the abstract, “We hypothesize that tight glycaemic control in pregnant women diagnosed with GDM will result in similar clinical outcomes to women just below the diagnostic thresholds.”

The authors improved the results section.

In the discussion section the authors should discuss in more detail the mechanism involved in the tight glycaemic targets reduced rates of large for gestational age and caesarean sections.

The following was added in the Discussion:

4.4. Clinical outcomes

Treated GDM patients had a relative risk of LGA of 0.37, reducing the absolute risk from 12.6 to 4.6%. The RCTs, ACHOIS20 and MFMU 21, reported a decrease in LGA rates from 22 to 13%, 13% to 7.1%, respectively. As our CBG targets were lower it was not surprising that our rates of foetal overgrowth were also lower. The Pedersen hypothesis that states that maternal hyperglycaemia results in foetal hyperinsulinemia, glucose utilization, and subsequent fetal adiposity24. The HAPO study demonstrated the continuous linear associations between maternal glycaemia and cord c-peptide levels and rates of LGA13. It follows that lowering maternal glycaemia would lower rates of LGA and tight maternal glycaemic control should result in lower rates of LGA. Our study further supports the Pedersen hypothesis by demonstrating that tight glycaemic targets lower rates of LGA, even when compared to patients just below the diagnostic thresholds.

The HAPO study demonstrated that both maternal BMI and glycaemia have strong independent associations with caesarean section rates28. However, the HAPO study could not conclude that there was a causal relationship between increased maternal BMI and caesarean sections rates28. Our subgroup analysis of patients based on BMI may suggest tight glycaemic targets in women with BMI <30 kg/m2 does not reduce the risk of caesarean sections. As caesarean section rates decreased in women with GDM with BMI ≥30 kg/m2, it is obesity coupled with maternal glycaemia that is possibly the important factor in determining rates of caesarean sections.